# Effect of Stability of Two-Dimensional (2D) Aminoethyl Methacrylate Perovskite Using Lead-Based Materials for Ammonia Gas Sensor Application

**DOI:** 10.3390/polym14091853

**Published:** 2022-04-30

**Authors:** Muhamad Yuzaini Azrai Mat Yunin, Norfatihah Mohd Adenam, Wan M. Khairul, Abdul Hafidz Yusoff, Hasyiya Karimah Adli

**Affiliations:** 1Faculty of Bioengineering and Technology, Universiti Malaysia Kelantan, Jeli 17600, Malaysia; azrai9795@gmail.com (M.Y.A.M.Y.); norfatihah.mohdadenam@gmail.com (N.M.A.); 2Advanced Nano Materials (ANoMa) Research Group, Faculty of Science and Marine Environment, Universiti Malaysia Terengganu, Kuala Nerus 21030, Malaysia; wmkhairul@umt.edu.my; 3Gold Rare Earth and Material Technopreneurship Centre (GREAT), Universiti Malaysia Kelantan, Jeli Campus, Jeli 17600, Malaysia; 4Institute for Artificial Intelligence and Big Data, Universiti Malaysia Kelantan, City Campus, Kota Bharu 16100, Malaysia; 5Department of Data Science, Universiti Malaysia Kelantan, City Campus, Kota Bharu 16100, Malaysia

**Keywords:** lead-based perovskite, one-step sequential reaction, material stability, gas sensor

## Abstract

Changes in physical properties of (H_2_C=C(CH_3_)CO_2_CH_2_CH_2_NH_3_)_2_PbI_2_Cl_2_ and (H_2_C=C(CH_3_)CO_2_CH_2_CH_2_NH_3_)_2_Pb(NO_3_)_2_Cl_2_ (2D) perovskite materials from iodide-based (*I*-AMP) and nitrate-based (*N*-AMP) leads were investigated at different durations (days) for various storage conditions. UV-Vis spectra of both samples showed an absorption band of around *λ*_max_ 420 nm due to the transition of *n* to π* of ethylene (C=C) and amine (NH_2_). XRD perovskite peaks could be observed at approximately 25.35° (*I*-AMP) and 23.1° (*N*-AMP). However, a major shift in *I*-AMP and dramatic changes in the crystallite size, FHWM and crystallinity percentage highlighted the instability of the iodide-based material. In contrast, *N*-AMP showed superior stability with 96.76% crystallinity even at D20 under the S condition. Both materials were exposed to ammonia (NH_3_) gas, and a new XRD peak of ammonium lead iodide (NH_4_PbI_3_) with a red-shifted perovskite peak (101) was observed for the case of *I*-AMP. Based on the FWHM, crystallite size, crystallinity and lattice strain analysis, it can be concluded *N*-AMP’s stability was maintained even after a few days of exposure to the said gases. These novel nitrate-based lead perovskite materials exhibited great potential for stable perovskite 2D materials and recorded less toxicity compared to famous lead iodide (PbI_2_) material.

## 1. Introduction

Perovskite materials have emerged as the most promising and efficient low-cost energy materials for various optoelectronic and photonic device applications [1]. Currently, perovskite materials are also being explored for use in laser cooling and photocatalysis [2,3]. In 2009, Miyasaka et al. first applied organic lead halide perovskite in liquid-dye-sensitized solar cells with a conversion efficiency of 3.8% [4]. Beginning with the breakthrough in solid-state perovskite-sensitized solar cells in 2012, various studies have been carried out on photovoltaic devices based on halide perovskite materials [5,6] due to their excellent optical and electrical characteristics [7]. The biggest advantages of solid-state perovskite are a direct band gap with a high molar extinction coefficient and low trap densities, which cause long-range free-carrier diffusion lengths (≈100 nm) [8,9]. It was later reported that perovskite materials have other unique physical properties, such as a high-absorption coefficient, long-range ambipolar charge transport, low exciton-binding energy, high dielectric constant and ferroelectric properties [10]. Due to the growing interest in perovskite materials, a variety of applications have been explored for solar cells, light-emitting devices, semiconductor devices, transistors waste management applications and gas sensor applications [11,12,13,14].

Despite the enormous potential of perovskite, which is typically three-dimensional (3D), the material possesses a brittle property, which has become the primary concern of most researchers. The material is prone to lattice degradation when exposed to moisture [12,15], oxygen or ultraviolet (UV) radiation [8]. The organic cation in perovskites, methylammonium (MA), was recognised as the main cause for its instability [16,17]. Perovskite sensitivity also depends on its structure, hydrophobicity of the alkyl chains, the chemical environment surrounding the heteroatom and stereochemistry [16]. Barbé and coworkers reported that hydrated-phase 3D perovskite of methylammonium lead iodide (MAPbI_3_) would form as a result of absorbing molecules of water at ~40% relative humidity (RH). It was also found that material conversion of perovskite can be accelerated when an electric field is applied to the material, resulting in irreversible phase degradation of MAPbI_3_ to lead iodide (PbI_2_) [18,19] and the aqueous form of methyl ammonium iodide (CH_3_NH_3_I) [20].

Tremendous efforts have been carried out to enhance the stability of 3D perovskite through various approaches, such as compositional engineering, interfacial regulation, defect passivation and device encapsulation [21,22]. In 1986, Maruyama et al. reported the first two-dimensional (2D) perovskite material that formed through self-assembly of layers held together by van der Waals and intermolecular forces. The introduction of long-chain hydrophobic organic cations of 2D perovskites ensured long-term stability and improved the performance simultaneously [23,24]. Two-dimensional congeners of well-known 3D perovskites display new properties enabled by their reduced dimensionality [25]. Reducing the dimensionality of 3D bulk perovskites has led to the formation of another class of perovskite materials known as layered perovskites [26,27]. Layered perovskite structures comprise an infinite number of 2D slabs with ABX_3_ structures separated by an element. In the case of organic–inorganic perovskites, the long organic cations in the octahedral structure, which do not fit the structure, act as a barrier and cap the 3D perovskite, resulting in the formation of a 2D structure. In addition, the organic cations with high ionic radii usually develop 2D structures [16]. Figure 1 highlights the comparison between conventional 3D and 2D perovskites structures [28].

In 2D perovskites, additional larger organic cations are introduced as spacers, isolating inorganic metallic octahedra layers to form quantum well superlattices [28]. The use of hydrophobic spacer cations efficiently separates the ionic lattice of inorganic octahedrons from ambient water molecules [29]. In addition, additional spacers and asymmetrical lattice designs allow further freedom to adjust inherent physical characteristics, including the optical band gap, exciting energy and dielectric constant [29,30]. Two-dimensional perovskite based on (RNH_3_)_2_A*_n_*
_− 1_M*_n_*X_3*n* + 1_ (Ruddlesden–Popper phase) wherein RNH_3_ is a primary aliphatic or aromatic alkylammonium cation acting as a spacer between the perovskite layers and A, M, X and *n* represent small organic cations (MA^+^, FA^+^ or Cs^+^), B-cations in 3D perovskite (Pb^2+^ and Sn^2+^), halide anions (I^−^, Br^−^ and Cl^−^) and the number of octahedrons in each individual perovskite layer reported previously [28,31,32]. Compared to 3D perovskite, which is composed of octahedral [MX_6_]^4−^, 2D perovskite is more akin to a sandwich structure, connecting different layers through large space cations. This type of perovskite has been widely reported for its improved photoelectric performance [33]. Phenyl ethyl ammonium iodide (PEAI), which was introduced as a bulky cation in 2D perovskite and fabricated in a layered 2D/3D structure, managed to achieve 20.1% power conversion efficiency (PCE) with 85% PCE retention after 800 h even under ambient conditions [34]. In addition, Bis(phenylethylammonium) bis(methylammonium) lead iodide, (PEA)_2_(MA)_2_Pb_3_I_10_ (PEA = C_6_H_5_(CH_2_)_2_NH_3_^+^, MA = CH_3_NH_3_^+^), showed no change after exposure to 52% relative humidity for 46 days, although MAPbI_3_ can be completely converted to lead iodide, PbI_2_ [35]. A similar result was reported for (BA)_2_(MA)_2_Pb_3_I_10_, which remained unchanged after 2 months of exposure under 40% RH [36].

However, previous studies have reported that titanium dioxide layers preserved adequate porosity with a small amount of physiosorbed water, which led to an increase in perovskite conversion from the starting materials [37,38]. In addition, surface coating with polymers, COFs or SiO_2_ could also enhance the stability [39,40]. According to Zhu et al., the covalent organic frameworks (PNCs–COFs) demonstrated good water processability originating from the hydrophobic COF’s shell. Furthermore, the COF’s coating enables the formation of the PNCs–COFs heterojunction, which facilitates the generation and transportation of photoinduced charge carriers [39]. Huang et al. have also reported that the inorganic material coatings of SiO_2_ are preferred for coupling with perovskites to improve their stability, whereas the conventional SiO_2_ formation method is unsuitable as it requires water [40]. The structural stability of perovskite is also characterized by the absence of polymorphism and the capacity to remain stable in a particular crystalline period under very broad circumstances, such as heat and pressure [41]. Two-dimensional perovskite becomes more stable than its 3D counterpart due to relaxation of the hydrogen bonds at the surface, which makes the degradation speed of 2D perovskites much slower than that of the 3D materials [42,43]. By focusing on the material itself, it will not influence the surface activity sites, the optical properties of materials for gas sensors or flexibility of the device fabrication.

The sensitivity of perovskite-based materials makes them potential candidates for sensor material. To date, few articles have reported halide-perovskite-based materials with tuned material characteristics from manufacturing procedures for ammonia gas sensor application [44]. Zhao et al. reported that ammonia gas induces a phase transformation of perovskite (CH_3_NH_3_)PbI_3_ film, leading to a rapid (<1 s) change in its colour from brown to colourless. This colour change is reversed within seconds upon removal of the ammonia gas, suggesting the potential use of perovskite halides for ammonia sensor applications [45]. Meanwhile, the inorganic 2D structure of NbWO_6_ perovskite was reported to exhibit fast hydrogen sulphide gas sensing of less than 6 s with high selectivity and sensitivity at 150 °C [46]. Passivating 3D perovskite films with a long alkyl chain organic cation of *n*-octylammonium bromide will form mixed-dimensional 2D/3D perovskite films. Performance and ambient stability of these perovskite-based NO_2_ gas sensors were greatly enhanced with a sensitivity of 6.3 ± 0.83 times per ppm and quick response/recovery times of 5.7 s and 12.7 s, respectively [47]. These findings highlight the potential of producing gas sensors using 2D perovskite with the advantages of a wider band gap and better stability in ambient circumstances.

Exploration of 2D perovskite by previous studies [47,48,49,50] motivated us to synthesize novel 2D aminoethyl methacrylate perovskites (H_2_C=C(CH_3_)CO_2_CH_2_CH_2_NH_3_)_2_PbI_2_Cl_2_ and (H_2_C=C(CH_3_)CO_2_CH_2_CH_2_NH_3_)_2_Pb(NO_3_)_2_Cl_2_ based on lead (ii) iodide and lead (ii) nitrate, as in Equation (1) and (2), as inorganic material precursors. The prepared perovskites were kept under different conditions, including room temperature (RT), silica (S) and vacuum (V), for a different number of days to compare the changes in physical properties and stability of the materials, and further to test the potential of the materials for ammonia gas sensing.
(1)$$\begin{array}{ccccc} 2H_{2}C=C(CH_{3})\;CO_{2}CH_{2}CH_{2}NH_{3}\cdot \mathrm{Cl} & + & PbI_{2} & ⟶ & (H_{2}C=C(CH_{3})CO_{2}CH_{2}CH_{2}NH_{3}{)}_{2}PbI_{2}Cl_{2} \end{array}$$
(2)$$\begin{array}{ccccc} 2H_{2}C=C(CH_{3})\;CO_{2}CH_{2}CH_{2}NH_{3}\cdot \mathrm{Cl} & + & Pb({NO_{3})}_{2} & ⟶ & (H_{2}C=C(CH_{3})CO_{2}CH_{2}CH_{2}NH_{3}{)}_{2}Pb({NO_{3})}_{2}Cl_{2} \end{array}$$

## 2. Materials and Methods

### 2.1. Materials

Lead (II) iodide (I) (Sigma-Aldrich (M) Sdn. Bhd., Selangor, Malaysia) (99%), lead (II) nitrate (N) (Sigma-Aldrich (M) Sdn. Bhd., Selangor, Malaysia) (≥99.0%) and 2-aminoethyl methacrylate hydrochloride (AEMA) (Acros Organics) (90%, stabilized) were the main chemicals that were used without further purification. Other chemicals used in this research include titanium (IV) dioxide (TiO_2_) powder (R&M chemicals) (≥99.5%, from 1 to 150 nm) and titanium diisopropoxide bis(acetylacetonate) (TAA) (Sigma-Aldrich (M) Sdn. Bhd., Selangor, Malaysia) (75 wt. % in isopropanol). This reaction employs a number of solvents, including *N,N*-Dimethylformamide (DMF) (Sigma-Aldrich (M) Sdn. Bhd., Selangor, Malaysia) (anhydrous, 99.8%), dimethyl sulfoxide (DMSO) (Sigma-Aldrich, Subang Jaya, Malaysia) (anhydrous, ≥99.9%), ethylene glycol (EG) (R&M chemicals, Subang, Malaysia), ethanol absolute (HmbG Reagent Chemicals, Malaysia) and deionized water.

### 2.2. Fabrication

An indium tin oxide (ITO) conductive plate was cut into 2 cm × 2 cm. ITO glasses (70–100 Ω/sq.) (Sigma-Aldrich) were cleaned using 10% Decon 90 (Decon Laboratories Limited, Hove, UK) detergent and rinsed with distilled water to remove any surface particles. Then, ITO glasses were sonicated in a water bath for 15 min at 50 °C. The cleansed ITO glasses were dipped in and rinsed with ethanol for 3 min to remove unwanted particles. Next was the drying process where the ITO glasses were dried and exposed under an ultraviolet lamp for 15 min before being stored in dark conditions [51,52].

For the titanium (IV) dioxide (TiO_2_) paste (*p*TiO_2_), 2 g of TiO_2_ powder was dissolved in 100 mL of ethylene glycol and stirred for 30 min. Once a homogeneous solution was obtained, the dilution process was carried out by mixing 1 mL of TiO_2_ paste into 100 mL of ethanol. The solution was stirred overnight under dark conditions using a magnetic stirrer before use. A dense TiO_2_ layer (*d*TiO_2_) was deposited on the ITO/glass substrate by spin-coating at 5000 rpm for 10 s using 4 mL of ethanoic solution containing 0.3 mL of titanium diisopropoxide bis(acetylacetonate) and annealing at a temperature of 500 °C for 35 min [12]. Then, the *p*TiO_2_ layer was deposited on the *d*TiO_2_-covered ITO/glass (*d*TiO_2_/ITO/glass) by spin-coating 50 µL of a solution of *p*TiO_2_ at 2000 rpm for 20 s followed by drying at 125°C and heat treatment at a temperature of 550 °C for 35 min to form *p*TiO_2_*/d*TiO_2_/ITO/glass. *p*TiO_2_/*d*TiO_2_ at 500 °C was preheated before use.

Initially, 0.42 M of **N** was prepared in 3 mL of a 2:1 (*v:v*) solvent mixture of ethylene glycol and deionized water and 0.83 M of AEMA in 3 mL ethylene glycol. For the synthesis of 2D aminoethyl methacrylate perovskite (AMP), the solution of N and AEMA were mixed and heated up to 100 °C. The prepared ITO glass that contained a TiO_2_ layer was then immersed in the mixed solution of N and AEMA for 30 min. Then, it was dynamically spin-coated at 1000 rpm for 10 s, followed by a second step at 5000 rpm for 30 s before being annealed at 200 °C for 30 min. The process was repeated with the film immersed again in the AEMA solution for 15 min and spin-coated at 1000 rpm for 10 s and 5000 rpm for 30 s. The perovskite thin film obtained was then annealed at 100 °C for 10 min and at 200 °C for 1 h and denoted as *N*-AMP. In a similar procedure, the synthesis of *I*-AMP was obtained from the mixture of 0.42 M of I in 3 mL of a 9:1 (*v:v)* solvent mixture of N,N-dimethylformamide (DMF) and dimethyl sulfoxide (DMSO) and 0.83 M of AEMA in 3 mL ethylene glycol. The final substrates of *N*-AMP and *I*-AMP are shown in Figure 2. The fabrication process of *I*-AMP and *N*-AMP is illustrated in Figure 3. Both samples were stored in different storage conditions, including room temperature (25 °C) (RT), silica (S) and vacuum (V), upon characterizations at day 1 (D1) (after 24 h of preparation), day 15 (D15) and day 20 (D20).

### 2.3. Characterization

Several physical characterizations were conducted to evaluate the changes in the materials at day 1 (D1) (after 24 h of preparation), day 15 (D15) and day 20 (D20) under different storage conditions, including room temperature (RT), silica (S) and vacuum (V), using an optical microscope (OM), ultraviolet-visible spectroscopy (UV-Vis) and X-ray diffraction (XRD) analyses. Microstructure morphologies of *I*-AMP and *N*-AMP were observed using ProgRes Microscope Cameras (Jenoptik, Jena, Germany) at 50× magnification. UV-Vis analysis was carried out using Spectroquant Pharo 300 UV-vis Spectrometer (Merck Sdn. Bhd., Selangor, Malaysia) to identify the optical properties of both samples with wavelengths in the range from 400 nm to 800 nm. From the UV-Vis spectrum, band gaps of the materials were calculated using a Tauc Plots graph of (*α**hv*)^(1/n)^ versus photon energy (*hv*). The crystallite size, crystalline structure and lattice strain were determined via X-ray diffraction (XRD D2 Phaser (Bruker, Shah Alam, Malaysia) under monochromated Cu K*α* irradiation (*λ* = 1.5418 Å) at a scan rate of 4 °C min^−1^ in the region from 10° to 90°.

For gas sensor application, *I*-AMP and *N*-AMP underwent direct exposure to 99.99% ammonia gas (Sani Sdn. Bhd.) at D1 for 5 min at a constant flow in a sealed vessel. Figure 4 shows the analysis set-up, which consisted of a custom-designed test chamber to control the gas environment. Before the analysis was performed, the prepared *I*-AMP and *N*-AMP were inserted individually into the test chamber, and the chamber was purged with 99.99% purified nitrogen gas (Dira Resources, Kota Bharu, Malaysia) for 15 min to eliminate any contamination. Gas flow to the chamber was organized by mass flow meter and controlled using Sierra’s SmartTrak, with N_2_ at 10 sccm (standard cubic centimetres per minute). After the purge, ammonia gas was then introduced into the chamber and the sample was exposed to the gas for 5 min. After the exposure, nitrogen gas was purged again for 10 min to push ammonia gas into another vessel that contains sulphuric acid as scrubber for gas ammonia neutralization (Figure 4). Changes in crystallite size, crystalline structure and lattice strain of *I*-AMP and *N*-AMP before and after ammonia gas exposure were investigated using an XRD spectrum.

## 3. Results and Discussion

### 3.1. Surface Morphology

Surface morphology observation on particle size and shape was performed using an optical microscope at 50× magnification under room temperature (RT), silica (S) and vacuum (V) at day 1 (D1), day 15 (D15) and day 20 (D20) as shown in Figure 5. For different durations under different preparation conditions, both samples (*I*-AMP and *N*-AMP) exhibited distinct characteristics. At D1, for *I*-AMP, the incomplete coverage of perovskite materials on the surface can be seen especially under RT conditions. Although the samples under S and V conditions revealed less exposure of pinhole, both images exhibited a needle-like crystal grain of lead iodide, PbI_2_, that was almost similar to the image reported by Burkitt et al. (2018) [53], which highlighted the appearance of small particles that dispersed in an irregular structure. Changes in surface morphology after several days can be expected when the degradation of perovskite starts to occur [54].

At D15, under RT, *I*-AMP was found to be distributed throughout the rougher surface. Disordered morphological features under RT could be attributed to the decomposition of crystalline perovskite [55]. Under V conditions, the irregular shape became more obvious, and the particles seemed connected and became isolated. At D20, under V conditions, almost full coverage of the perovskite material with a crystal grain of PbI_2_ could be seen on the surface. Poor coverage on the substrate was found as the formation of the islands was observed, hinting the degradation of perovskite occurred under RT and S conditions [56]. This can be explained by the fact that excess PbI_2_ can accelerate degradation, especially in the presence of oxygen, humidity, heat and light [57,58,59,60].

The surface morphology of *N*-AMP materials showed different patterns compared to the *I*-AMP material. Almost full coverage of the *N*-AMP material was found for all samples under all conditions. Under RT, the *I*-AMP material showed poor coverage, but *N*-AMP material images showed the opposite trend. Under RT and S, at D15, the morphology exhibited a honeycomb-like surface with interconnected pore channels [61,62]. As *N*-AMP was synthesized in a water and ethylene glycol mixture, the formed perovskite was found to be insensitive to water moisture from the ambient temperature. The surface morphologies of *I*-AMP and *N*-AMP under different conditions at D1, D15 and D20 are shown in Figure 5.

### 3.2. Optical Properties

UV–visible spectroscopy is a quantitative technique used to measure the ability of chemical substances to absorb light for multiple sample types, including liquids, solids, thin films and glass [63,64]. Molecules or parts of molecules that absorb light strongly in the UV-vis region are called chromophores [65]. In the UV-visible region, electron transitions occur when chromophores absorb UV light. The electronic transition is usually investigated in order to study the conjugated π systems of chromophores [66]. The UV light absorption caused excitation of outer electrons from their ground state to an excited state [67]. In this study, UV-Vis was used to show light absorption properties of *I*-AMP and *N*-AMP at ambient temperature from day 1 (D1) to day 20 (D20). Optical properties, such as electronic absorption and band gap energy, were calculated based on a UV-Vis spectrum in the range of wavelength from 400 nm to 800 nm. Absorption spectra of *I*-AMP and *N*-AMP solutions are shown in Figure 6.

Electronic absorption spectra of *I*-AMP and *N*-AMP may arise from ethylene (C=C) and amine (NH_2_) [68]. From Figure 6, the intensity band for both samples seen from around *λ*_max_ 400 nm to 420 nm could be attributed to the transition of *n* to π* [69]. Transitions of *n* to π* are mainly localized to the amine (NH_2_) chromophore [68] due to the excitation of the outer electron to the π* orbital. It can be seen that for all durations (D1, D15, D20), a bathochromic or red shift occurred in which the band position in a spectrum was moved to a longer wavelength. Browne and coworkers, in 2019, reported that the wavelength of light absorbed by the chromophore is influenced by how conjugated the molecule is [70]. An increase in the number of conjugations in the chromophore will cause a bathochromic shift. This shift causes a change in the energy levels of molecular orbitals based on the number of conjugations [69]. The existence of an amino group in 2-aminoethyl methacrylate hydrochloride as a chromophore (AMP) means that the nitrogen molecule has a lone pair that causes the bathochromic shift.

A band gap is the distance between the valence band and the conduction band of electrons. Essentially, the band gap represents the minimum energy that is required to excite an electron up to a state in the conduction band where it can participate in conduction [71,72]. An absorption edge of semiconductors corresponds to the threshold of charge transition between the highest nearly filled band and the lowest nearly empty band [73]. The band gap energy of a semiconductor describes the energy needed to excite an electron from the valence band to the conduction band. Determination of the band gap energy is crucial to identify photophysical and photochemical properties of semiconductors [74]. The optical band gap energy is determined based on the Tauc equation, as seen in Equation (3), proposed by Tauc in 1966 [74,75] where the optical band gap was obtained from the respective tangents to x-axis when y is equal to 0 [76,77]. According to the interband absorption theory, the optical band gap can be calculated using the following relation [78]:(3)$${\left(\alpha hv\right)}^{n}=A\left(hv-E_{g}\right)$$
where *α* is the absorption coefficient, *h* is Planck’s constant (6.63 × 10^−34^ J Hz^−1^), *v* is the frequency, *n* is the transition coefficient, which is equal to 2 (for allowed indirect transition) [79], *E*_g_ is the optical band gap of the materials, and *A* is a constant value.

The interband absorption process involves the transition of electrons between bands of solid materials. The absorption edge is caused by the initiation of optical transitions across the fundamental band gap [80,81,82]. Figure 7 shows the band gap energy values of *I*-AMP and *N*-AMP obtained from the plots (*αhv*)^2^ vs. *hv* for both samples at D1, D15 and D20, as shown in Appendix A.

From Figure 7, it can be observed that there was a slight decrease in band gap energy over time in the case of *I*-AMP. This was due to the decreasing trend in the crystallite size, correlated with XRD data, as the crystallite size of the materials is predominantly influenced by the band gap energy [77,83]. Thus, it can be concluded that the stability of *I*-AMP deteriorates over time, whilst the trend of the band gap energy of *N*-AMP shows the opposite trend. The increase in band gap energy over time was tallies with the increase in the crystallite size (XRD).

### 3.3. Structural and Crystallinity

Structural properties of perovskite film were analyzed by X-ray diffraction (XRD) analysis. By comparing the spectrum of the final product (*I*-AMP) with that of the starting materials (PbI_2_ and AEMA), as shown as Appendix A, the diffraction peak that appeared at 25.35⁰ corresponded to the (101) lattice planes of inorganic perovskite [84,85,86]. Other diffraction peaks at 12.5° (001) and 38.8° (004) represented PbI_2_ crystal grains [84] and titanium dioxide (TiO_2_), respectively [87,88]. Structural properties of *I*-AMP were then investigated at various durations, including at day 1 (D1), day 15 (D15) and day 20 (D20) and different storage conditions, including room temperature (RT), silica (S) and vacuum (V), as shown in Figure 8. From Figure 8, it can be seen that gradual decreases in perovskite peaks (101) (♦-marked peaks) corresponded to a gradual increase in PbI_2_ peaks (001) (▲-marked peaks) after a few days of preparation at all storage conditions. The appearance of the PbI_2_ peak around 12.5° can be expected because of the solution of perovskite in the mixture of DMSO and DMF, and a small amount of residual yellow PbI_2_ crystals are typically obtained in perovskite solutions and films [89,90]. Under RT conditions, after 24 h of preparation (D1), PbI_2_ peak intensity was found to be higher than that of perovskite, indicating rapid transformation of perovskite into a PbI_2_ by-product. This became obvious at D15 and D20 with the decrease in the intensity of the perovskite peak (101) and increase in the PbI_2_ peak (001) intensity throughout the days. The shift occurred mainly due to strain–stress present in the lattice [91,92]. Compared to other storage conditions, there was a red shift of peak location of *I*-AMP (101), although there was no significant change in PbI_2_ peak (001) intensity over the duration of the study. The peak intensity of the perovskite plane (101) was found to decrease with the increase in time. These phenomena indicated that the crystallographic perovskite structure was changed due to moisture-induced decomposition. As perovskite crystal decomposition leads to PbI_2_ conversion, it could be seen that the PbI_2_ peak (001) was higher with the decrease in the perovskite (101) peak (Figure 8). Overall, *I*-AMP showed high crystallinity under RT conditions while S and V conditions exhibited amorphous and broaden spectra.

Meanwhile, for the case of *N*-AMP (Appendix A), which contained lead nitrate and Pb(NO_3_)_2_ as the starting material, all samples exhibited main peaks at 23.1° (101) and Pb(NO_3_)_2_ at 19.3° (111), 22.4° (200) and 32.4° (220) in similar peak locations to those reported by other studies [90,93,94]. The conversion of Pb(NO_3_)_2_ to MAPbI_3_ was also confirmed by previous studies [95,96,97]. From Figure 9, it can be observed that there was only a slight shift in AMP peak location at 23.1° for all storage conditions. It can be said that there was only a small change in crystallite size and lattice strain [98]. Thus, it can be claimed that over time, *N*-AMP is more stable compared to *I*-AMP. The exceptionally long electron lifetime of Pb(NO_3_)_2_/water-based material can be explained by a benign defect inactivation effect, which arises from water (H_2_O) and nitrate ion (NO_3_^-^) residues in the aqueous precursor solution involved in the formation of perovskite crystals [99]. In addition, Pb(NO_3_)_2_ materials have interesting supramolecular chemistry because of the coordination of nitrogen and oxygen electrons in donating compounds to water molecules that have effective solubility properties and a bidentate nature, which makes it remain stable [100,101]. It was confirmed by previous studies that perovskite from Pb(NO_3_)_2_ materials is expected to exhibit a lower degradation rate compared to PbI_2_-based perovskite materials because of their bigger grain sizes, which result in improved stability [93]. These large-size Pb(NO_3_)_2_/water-based perovskite grains indicate that Pb(NO_3_)_2_ residue acts as an effective moisture scavenger [93,99].

However, the stability of *I*-AMP deteriorated over time as most organometallic perovskite from the halide group would decompose into hydroiodic acid (HI), methylamine (CH_3_NH_2_) and PbI_2_ in the presence of water moisture (H_2_O) or oxygen (O_2_) as a catalyst [102]. When perovskite is exposed to moisture and oxygen, its crystal structure reverts to an intermediate hydrate (CH_3_NH_3_)_4_PbI_6_.2H_2_O, which accelerates degradation [103,104]. According to Silva Filho and coworkers, it is well-known that conversion of PbI_2_ into perovskite leaves a small percentage of unconverted PbI_2_, which is attributed to quick growth of a thick shell of perovskite that hinders total conversion [105,106]. Therefore, based on the XRD results in Figure 8, it was found that the decomposition of *I*-AMP to PbI_2_ was noticeable through the PbI_2_ peak.

In order to investigate the change in perovskite material properties at various durations in different conditions, structural parameters of samples *I*-AMP and *N*-AMP were calculated (Appendix A). Figure 10 shows the full width at half maximum (FWHM), crystallite size, crystallinity and lattice strain of *I*-AMP (⸻) and N-AMP (∙∙∙∙∙). The FWHM was used to characterize material properties and surface condition features based on XRD peaks to calculate the crystallite size and lattice strain using Scherrer’s equation [107,108,109], as stated in Equations (4) and (5) as below:(4)$$D=\frac{k\lambda }{\beta cos\theta }$$
(5)$$\varepsilon =\frac{\beta cos\theta }{4}$$
where *D* is the average crystallite size (nm), *k* is the Scherrer constant (0.89), *λ* is the wavelength of the incident X-rays (0.154 nm for Cu K*α* radiation), *β* is the full width at half-maximum of the reflection peak with the same maximum intensity in the diffraction pattern (in radians), θ is the Bragg diffraction angle (in degrees), and ε is the lattice strain (as a percentage).

From Figure 10a, it can be observed that FWHM values of *I*-AMP increased over time under all conditions while *N*-AMP showed the opposite trend. The FHWMs were calculated at 2*θ* = 25.35° (RT) and 29.0° (S and V) for *I*-AMP and at 2*θ* = 23.0° (RT, S and V) for *N*-AMP. A linear increase in the FHWM was reported due to the density of point defects, which affect crystallinity, crystallite size and lattice strain [110]. This indicates that peak width (FWHM) changes inversely with crystallite size, whereby as crystallite size decreases, the peak becomes broader and exhibits the amorphous structure of the XRD spectrum. Crystallite size broadening is most pronounced at large angles of 2*θ* at around 28.5°–29.8°. It can be concluded that if the values of the FHWM increase, the crystallite size of perovskite will decrease.

Crystallite size is an important parameter to determine whether the material is soft (small crystallites) or brittle (large crystallites), as well as the thermal and diffusion behaviour of semicrystalline [111]. Crystallite size generally corresponds to the coherent volume, size of the grains of a powder sample, thickness of polycrystalline thin films or bulk material at respective diffraction peaks [112]. Changes in crystallite size can be realized due to the increase in dislocation density and microstrain. Guo et al. stated that perovskite degradation causes phase segregation, morphological deformation and lattice shrinkage, which could be contributed to crystallite size reduction [113]. As expected, a decreasing trend of the crystallite size of *I*-AMP was found as FHWM values increased, with *N*-AMP trends showing the opposite (Figure 10b). The crystallite size of *I*-AMP under all conditions became smaller over the course of time from 29.49 to 22.54 nm from D1 to D20 for samples kept under RT conditions. Under RT conditions, the decrease in crystallite size may be due to some reactions with residual moisture trapped in the structure, most likely derived from water dispersion, leading to decomposition of the perovskite [114]. The decline in the crystallite size of *I*-AMP may also be due to decomposition of perovskite into PbI_2_, which leads to point and interstitial defects and/or vacancy in the perovskite lattice, which causes dislocation of atoms in the structure [115]. In contrast, the crystallite size of *N*-AMP increased over time under all conditions. An increase in crystallite size will affected grain size as it consists of agglomerations’ crystallite size, whereby increasing the crystallinity of the material leads to an increase in crystallite size and grain size [116]. In comparison, the crystallite size of *N*-AMP was bigger than *I*-AMP even at D1. The decrease in crystallite size leads to gradual degradation of the perovskite material [77]. Thus, it can be concluded that *I*-AMP was less stable compared to *N*-AMP with a decrease in crystallite size over the course of time from 29.49 to 22.54 nm for RT and 1.27 to 0.82 nm for S and 1.28 to 0.95 nm for V.

In order to find the ratio of intensity from crystalline peaks to the sum of the crystalline and amorphous intensities, the crystallinity percentage was calculated based on Equation (6):(6)$$I_{o}=\frac{I_{crystalline}}{I_{total}}\times 100$$
where $\;I_{o}$ is the crystallinity percent (%), $\;I_{crystalline}$ is the area of crystalline peaks and $I_{total}$ is total area of peaks.

As shown in Figure 10c, at D1, the crystallinity of *I*-AMP recorded the highest value under RT conditions at 63.57%, compared to 38.35% and 24.84% for S and V conditions. However, over the course of time until D20, the crystallinity of *I*-AMP experienced a sharp decline in which the percentage of crystallinity decreased from 63.57% to 40.50% for RT, 38.35% to 26.77% for S and 24.84% to 19.31% for V. It is known that crystallinity is related to crystallite size, whereby a smaller crystallite size exhibits less crystallinity [77]. The high crystallinity of *I*-AMP under RT conditions at D1 might be due to the existence of water moisture at RT, which can induce the crystallinity of perovskite material [117], answering the question of why crystallinity decreased over time under S and V conditions. Based on the literature, a moderate amount of water can facilitate nucleation and crystallization of perovskite material, resulting in better perovskite film quality and better device stability [118,119]. Under different conditions, crystallinity values are rather small and tally with the amorphous structure observed in the XRD spectrum (Figure 8). A decrease in crystallinity over time was observed for all samples of *I*-AMP, proving the occurrence of perovskite degradation [77]. In contrast, *N*-AMP showed an increasing value of crystallinity from D1 to D20 under all conditions, proving that perovskite remained stable over time (Figure 10c), with the highest percentage of crystallinity being 96.76% at D20 in S conditions. The stability of *I*-AMP and *N*-AMP could also be related to the lead-based starting material used for perovskite preparation. PbI_2_ of *I*-AMP was reported to be soluble only in hot water, while Pb(NO_3_)_2_ of *N*-AMP is highly soluble in water. Therefore, water may react first with Pb(NO_3_)_2_, which acts as a water scavenger, retarding accelerated decomposition and enhanced material stability [93].

In particular, the lattice strain indicated that atoms of the material were displaced from their reference lattice position. Dislocation density exhibits crystallographic imperfection in a film induced by interstitial atoms and/or vacancies [115]. Effects of crystallite size on lattice strain and crystal structure can be explained from Rietveld refinement and the Williamson–Hall plot, respectively [120]. Several studies reported that the structural lattice parameters decrease with increases in crystallite size [77,120]. Hence, the increase in the trend of lattice strain (Figure 10d) of *I*-AMP for all conditions from D1 to D20 correlated with the decrease in crystallite size. The same pattern was observed for *N*-AMP, whereby the lattice strain was found to decrease gradually after a few days when there was an increase in crystallite size. Lattice strain is used as reference for the measurement of lattice constant distribution, such as lattice dislocations that arise from crystal imperfection [121]. Lattice strain in the crystals possibly generated during crystallization contributed to the change of lattice parameters, such as unit cell constants, unit cell volume, bond lengths and tolerance factor [90,120]. Islam et al. reported that the perovskite decomposition process leads to an increase in crystal defects, including point defects, edge dislocation and precipitates formed [115]. It is well-known that the point defects lead to lattice misfit and dislocation in the film structure [122]. Thus, it can be concluded that *N*-AMP is more stable than *I*-AMP, and over the course of time, *N*-AMP showed a lack of defects in the film structure because its lattice strain showed decreasing trends with the lowest value recorded at 0.06% for D20 under S conditions.

### 3.4. Effect of Ammonia Exposure on the Structure and Crystallinity

Several factors govern the interaction between ammonia gas molecules and perovskite compounds, namely the type of perovskite material, microstructure, ambient conditions, time exposure, pressure and light intensity conditions [123]. In order to detect any structural change in the AMP film in the presence of ammonia (NH_3_), X-ray diffraction (XRD) analysis was performed on NH_3_ exposed to *I*-AMP and *N*-AMP films. Figure 11 shows the comparison of XRD spectra of *I*-AMP before and after gas exposure, denoted as A and B, respectively. 

It can be seen that the peak positions of *I*-AMP are incomparable to the recorded spectrum before exposure. Interestingly, an unknown broad peak was observed around 24.2° (Figure 11B). Previous research reported that when MAPbI_3_ material is continuously exposed to ammonia gas, amino groups (NH^+3^) will interact with the perovskite material [123], which will most likely form ammonium lead halide with water, NH_4_PbI_3_-H_2_O [124], ammonium lead halide, NH_4_PbI_3_ [125] and ammonium lead halide with methylamine, NH_4_PbI_3_-CH_3_NH_2_ [126]. In this case, the new phase peak formed was identified in Figure 11 as NH_4_PbI_3_ based on published data [127,128], which indicates a complete breakdown and degradation of *I*-AMP. It is noted that after exposure to NH_3_ gas, diffraction patterns of the NH_4_PbI_3_ phase started to appear along with *I*-AMP, indicating the same proton transfer reaction. According to Sasmal et al., proton exchange occurs between ammonium salt and NH_3_ molecules during the exposure, which leads to the formation of free amines [124]. Hence, during the transformation of *I*-AMP to NH_4_PbI_3_, there is an exchange of cations in the presence of NH_3_ gas molecules with the transfer of an electron from an ammonia molecule to a nitrogen-bound hydrogen atom. The characteristic XRD peak at 12.5° for PbI_2_ was found to be indistinguishable from the background signal after the exposure. It can be understood that there was an irreversible change in the crystal structure of the *I*-AMP phase after the removal of ammonia. This demonstrates that the perovskite was unable to form the original phase because it was stored under ambient conditions, whilst the diffraction peak of perovskite (101) at 25.35° shifted to red at 26.75° after gas exposure, and the spectrum turned into an amorphous structure. This indicates dramatic changes in the structural properties of the perovskite due to interaction with NH_3_ at room temperature [126].

Meanwhile, for the case of *N*-AMP (Figure 12), it can be observed that there was only a slight shift of the AMP peak from 23.17° (A) to 23.15° (B). However, it is worth noting that the enhancement of intensity occurred for all peaks. Sheikh et al. reported that when a higher concentration of ammonia is used, diffusion and interaction of ammonia gas molecules with perovskite will enhance and lead to an increase in the electrical conductivity of perovskite substrates [123]. Surface morphology also plays an important role in good application of gas sensors [129,130,131]. The porous surface of *N*-AMP (as in Figure 5) should provide a high surface area for NH_3_ molecule interaction, as well as active surface sites and diffusion channels for improved NH_3_ molecule interaction [123]. When ammonia molecules come in contact with the perovskite surface, they enter into the voids of perovskite grains and donate an electron, enhancing electrical conductivity [123]. For the case of *N*-AMP, no colour change was observed during the experiments, which were run at a low concentration (10 ppm) of NH_3_ for 5 min. At low exposure, the interaction of ammonia gas molecules might only occur on the outermost surface of *N*-AMP perovskite grains.

Changes experienced by *I*-AMP and *N*-AMP perovskite materials before and after gas exposure were further evaluated based on the full width at half maximum (FWHM), crystallite size, crystallinity and lattice strain, as shown in Figure 13. A summary of parameters is provided in Appendix A.

From Figure 13a, it can be observed that FWHM values of *I*-AMP increased after exposure while those of *N*-AMP showed the opposite trend. The FHWM for ammonia gas exposure was calculated at 2*θ* = 26.75° for *I*-AMP and at 2*θ* = 23.15° for *N*-AMP. As FHWM increased, the crystallite size of perovskite decreased (Figure 13b), during which the values dropped from 29.49 to 1.02 nm for the case of *I*-AMP. This reduced crystallite size could be due to perovskite deterioration caused by phase segregation, morphological deformation and lattice shrinkage [57]. In contrast, the crystallite size of *N*-AMP increased, affecting the grain size and crystallinity since it is made up of agglomerations of crystallite [116]. As shown in Figure 13c, the crystallinity of *I*-AMP decreased from 63.57% to 16.02% after exposure. Previous studies reported that structural lattice parameters decrease with increased crystallite size [77,120]. Hence, the increase in the trend of *I*-AMP lattice strain (Figure 13d) after exposure was correlated to the decrease in crystallite size. The same pattern was observed for *N*-AMP, whereby the lattice strain was found to decrease.

In conclusion, the stability of *N*-AMP was found to be superior compared to *I*-AMP even after exposure. Because ammonia molecules have stronger polarity than MA, NH_4_^+^ will easily replace MA^+^, resulting in decomposition and phase transformation of perovskite and an irreversible change in the colour of the device [132]. According to previous reports, changes in the lattice structure of lead (ii)-iodide-based perovskite with NH_3_ is due to the voids in the crystal lattice of perovskites that make the structure more vulnerable to NH_3_ molecules and are easily ruptured [133,134]. Thus, relatively close-packed crystal lattices of perovskite may reduce penetration of NH_3_ molecules [124]. Overall, the results suggest that 2D-layered hybrid perovskites based on long-chain alkyl ammoniums are able to improve the stability of NH_3_ sensing by using lead (ii) nitrate as a precursor compared lead (ii) iodide, indicating a significant advantage of robust perovskite substrates with extraordinary stability for ammonia sensing.

## 4. Conclusions

This study successfully synthesized lead-iodide-based and lead-nitrate-based two-dimensional (2D) aminoethyl methacrylate perovskites in a one-step sequential reaction for ammonia gas sensor application. Changes in physical properties, the stability of materials and gas performance were identified and compared for various days from day 1 (D1) to day 20 (D20) in different storage conditions, including room temperature (RT), silica (S) and vacuum (V). Distinct surface morphology characteristics were exhibited by *I*-AMP and *N*-AMP. For *I*-AMP, incomplete coverage of perovskite materials on the surface could be seen, especially at RT, and surface morphology changes were identified over time. In contrast, the *N*-AMP material showed almost full coverage of perovskite materials under all conditions. Electronic absorption spectra of *I*-AMP and *N*-AMP exhibited a band from *λ*_max_ 400 nm to 420 nm, which can be attributed to the transition of *n* to π* from ethylene (C=C) and amine (NH_2_). By comparing the XRD spectrum of the final product (*I*-AMP) with the starting materials (PbI_2_ and AEMA), the diffraction peak that appeared at 25.35° corresponded to the (101) lattice planes of inorganic perovskite. Meanwhile, *N*-AMP exhibited main peaks of perovskite at 23.1° (101) and Pb(NO_3_)_2_ at 19.3° (111), 22.4° (200) and 32.4° (220). The decreases in crystallite size, crystallinity and band gap with time confirm the decrease in the stability of *I*-AMP materials, whilst for *N*-AMP, the results show the opposite trend—crystallite size, crystallinity and band gap energy revealed increasing trends over time, hinting at the superior stability of the material. The stability of the materials was then tested via exposure to ammonia (NH_3_) gas. In order to detect any structural changes in the AMP film in the presence of ammonia (NH_3_), XRD analysis was performed for *I*-AMP and *N*-AMP films. Formation of the NH_4_PbI_3_ phase peak was identified from the *I*-AMP spectrum, which indicates a complete breakdown and degradation of *I*-AMP. In addition, the diffraction peak of perovskite (101) at 25.35° was red-shifted after gas exposure, and the spectrum developed an amorphous structure, indicating dramatic changes in structural properties of the perovskite material. Meanwhile, for the case of *N*-AMP, there was only a slight shift in the peak located around 23°. As the FHWM increased, the crystallite size of *I*-AMP perovskite decreased, in contrast to *N*-AMP. The crystallinity of *I*-AMP was also found to decrease after exposure. In conclusion, the stability of *N*-AMP was found to be superior compared to that of *I*-AMP even after ammonia gas exposure. Further research on the material stability of lead-based 2D perovskite materials is being undertaken for gas sensor application. In order to synthesize a stable hybrid perovskite for gas sensing application, the mechanistic pathway of complete degradation associated with irreversible reactions of AMP needs to be investigated in depth.

## Figures and Tables

**Figure 1 polymers-14-01853-f001:**
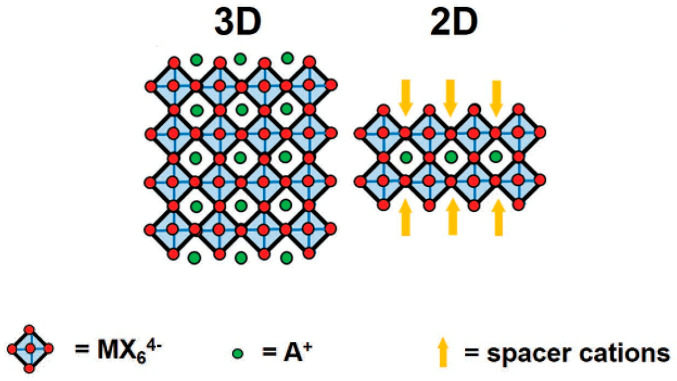
Structure of 3D (**left**) and 2D (**right**) perovskites, reproduced with permission from [28].

**Figure 2 polymers-14-01853-f002:**
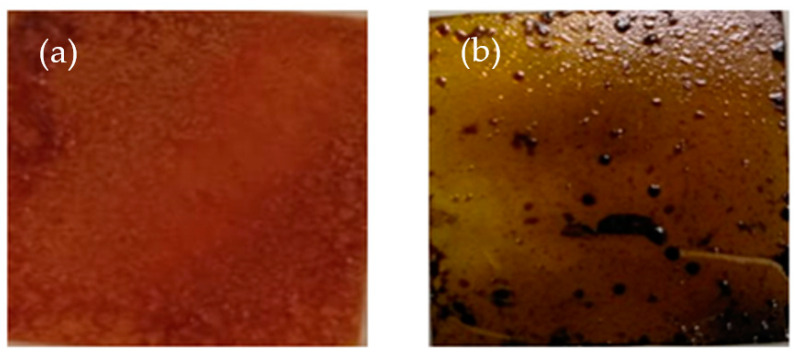
Final obtained substrates of (**a**) *N*-AMP and (**b**) *I*-AMP.

**Figure 3 polymers-14-01853-f003:**
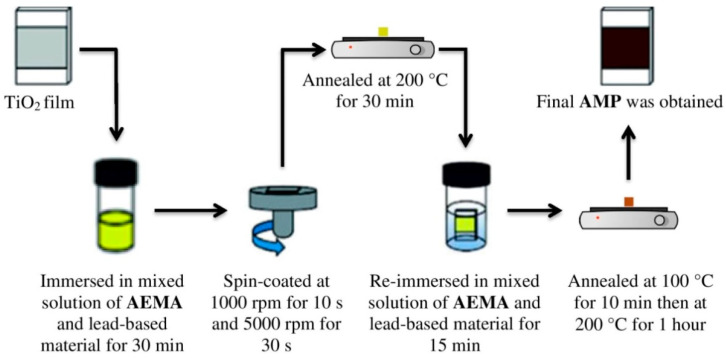
Synthesis of AMP from 2-Aminoethyl methacrylate hydrochloride (AEMA) and lead-based materials (I or N), respectively.

**Figure 4 polymers-14-01853-f004:**
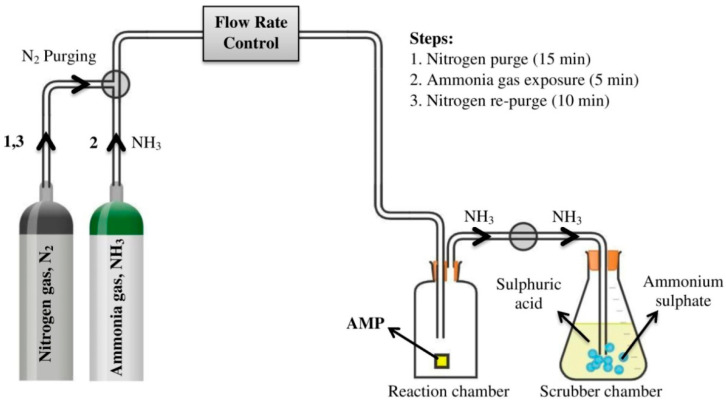
Process flow diagram for ammonia gas sensor application.

**Figure 5 polymers-14-01853-f005:**
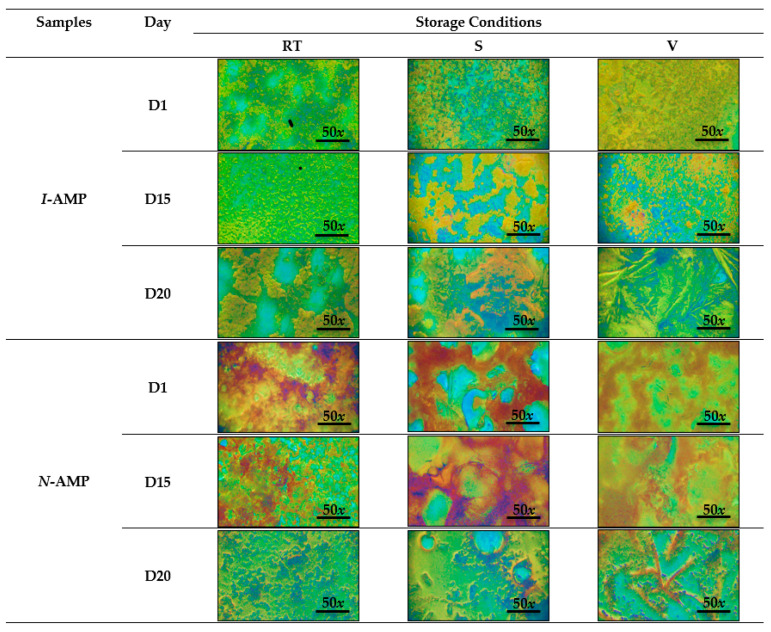
Optical microscopy images of *I*-AMP and *N*-AMP at room temperature (RT), silica (S) and vacuum (V) for D1, D15 and D20.

**Figure 6 polymers-14-01853-f006:**
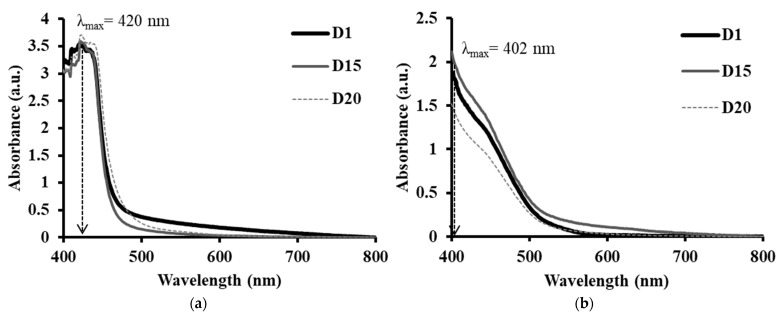
UV-Vis spectra of (**a**) *I*-AMP and (**b**) *N*-AMP for D1, D15 and D20, respectively.

**Figure 7 polymers-14-01853-f007:**
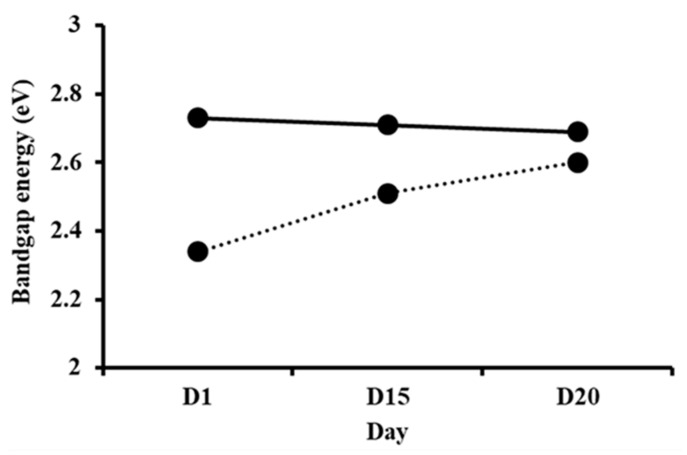
Band gap values of *I*-AMP (⸻) and *N*-AMP (∙∙∙∙∙) at D1, D15 and D20.

**Figure 8 polymers-14-01853-f008:**
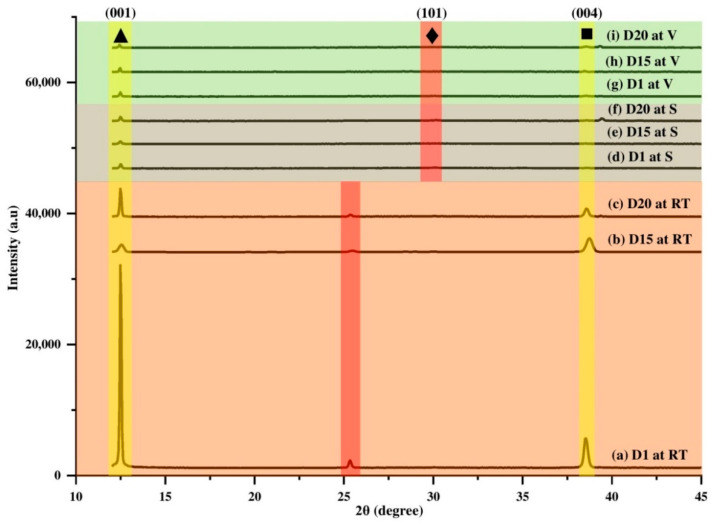
XRD spectrum of *I*-AMP shows ▲, ♦ and ■ marked for PbI_2_ (001), perovskite (101) and TiO_2_ (004), respectively.

**Figure 9 polymers-14-01853-f009:**
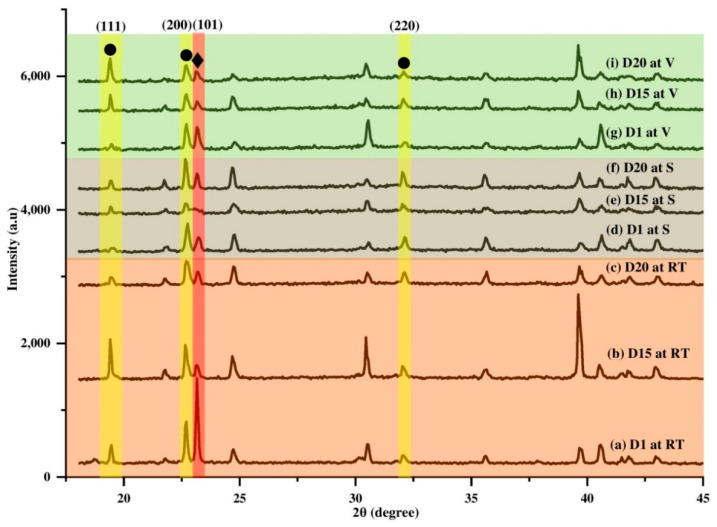
XRD spectrum of *N*-AMP with ♦ marking the peak of perovskite (101) and ● marking those of Pb(NO_3_)_2_ (111) (200) (220).

**Figure 10 polymers-14-01853-f010:**
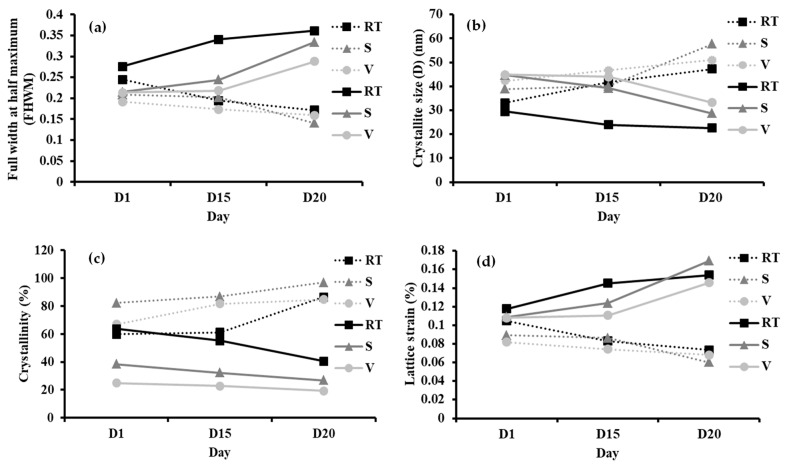
Parameters of: (**a**) full width at half maximum (FHWM), (**b**) crystallite size, (**c**) crystallinity and (**d**) lattice strain of *I*-AMP (⸻) and *N*-AMP (∙∙∙∙∙) with ■ marking RT, ▲ marking S and ● marking V.

**Figure 11 polymers-14-01853-f011:**
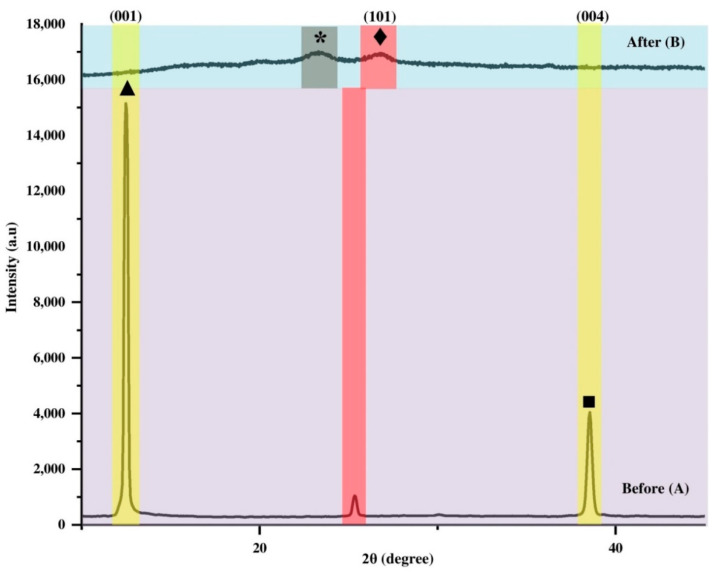
XRD spectrum of *I*-AMP with ▲, ♦, ■ and ⁎ representing PbI_2_ (001), perovskite (101), TiO_2_ (004) and new phase peak recorded before (**A**) and after exposure (**B**).

**Figure 12 polymers-14-01853-f012:**
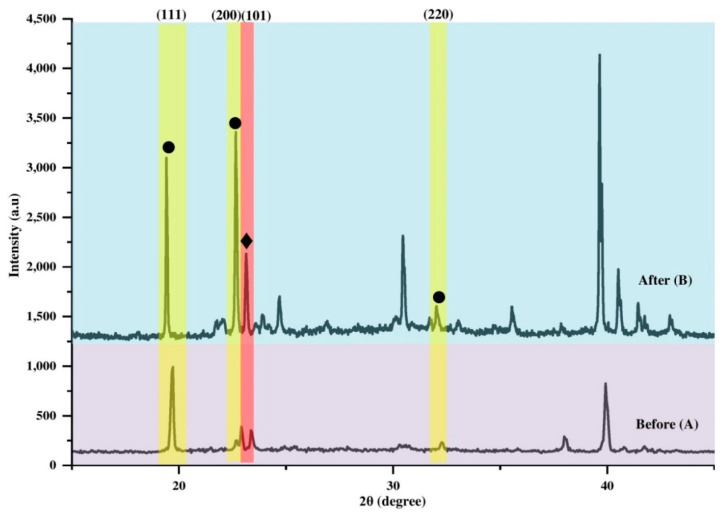
XRD spectrum of *N*-AMP with ♦ marking peak of perovskite (101) and ● marking those of Pb(NO_3_)_2_ (111) (200) (220), before (**A**) and after exposure (**B**).

**Figure 13 polymers-14-01853-f013:**
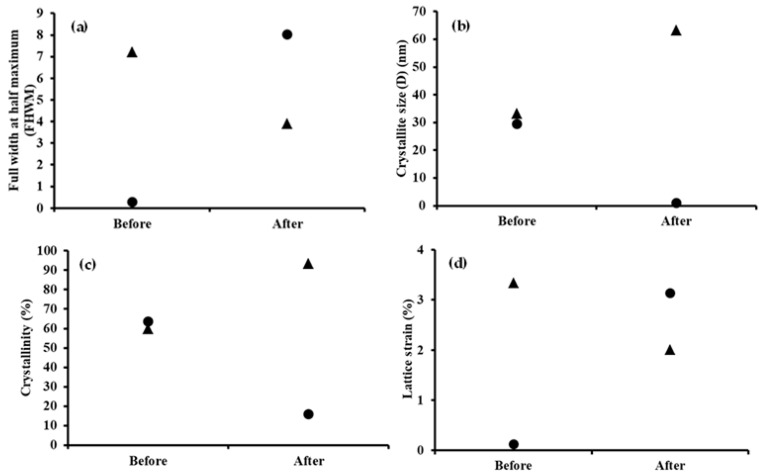
Parameters of: (**a**) FHWM, (**b**) crystallite size, (**c**) crystallinity and (**d**) lattice strain, before and after exposure (noted by the marks for *I*-AMP (●) and *N*-AMP (▲), respectively).

## Data Availability

The data presented in this study are available within the present article.

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
