# Peer review of "Effect of Stability of Two-Dimensional (2D) Aminoethyl Methacrylate Perovskite Using Lead-Based Materials for Ammonia Gas Sensor Application"

_polymers, 2022, doi:10.3390/polym14091853_

Round 1

Reviewer 1 Report

In this manuscript, the authors synthesized two-dimensional (2D) perovskites from aminoethyl methacrylate based on lead iodide and lead nitrate and compared the physical properties at different times under different storage conditions. As a conclusion, stability of N-AMP was found to be superior compared to I-AMP even after ammonia gas exposure. The research introduces additional information on the possibility of stabilizing the perovskite phases. In conclusion, I believe this article is suitable for publication in Polymers, with no changes to the manuscript.

Author Response

Thank you very much for the paper review. 

Reviewer 2 Report

Work very interesting and perspective for applications in practice and should be published. At the end, I have no attempts to produce solar cells on the basis of perovskite analyzed at work. It is known that not always the results obtained for individual materials overlap with the results for constructed devices. If you can ask for a comment at work in reference to the sentence in Conclusion:
Further Research on material Stability of Lead-Based 2D Perovskite
Materials Should Be Done Due to Their Advantages in Many Applications Such As Gas Sensor.

Figure 2 is only one compound. Is it corrected? 

What are the continued plans of the authors in the development of themes. Ask for a comment in Conclusion. If in relation to the development of Experimental methods, I recommend articles: Mater. Adv., 2022, 3, 2697
J. Phys. Chem. Lett., 2016, 7, 3458-3466.

Author Response

Chemical structure in Figure 2 is replaced with chemical equation for both peroskite materials, which are (H2C=C(CH3)CO2CH2CH2NH3)2PbI2Cl2 (I-AMP) and (H2C=C(CH3)CO2CH2CH2NH3)2Pb(NO3)2Cl2 (N-AMP) – page no. 4

A statement was revised in conclusion:

Further research on material stability of lead-based 2D perovskite materials is undertaking for gas sensor application. (Page no. 17)

The references were cited in experimental part:

·     Ratajczyk, P., Katrusiak, A., Bogdanowicz, K. A., Przybył, W., Krysiak, P., Kwak, A., & Iwan, A. (2022). Mechanical strain, thermal and pressure effects on the absorption edge of an organic charge-transfer polymer for flexible photovoltaics and sensors.  Mater. Adv., 2022, 3, 2697.

·     Szafranski, M., & Katrusiak, A. (2016). Mechanism of pressure-induced phase transitions, amorphization, and absorption-edge shift in photovoltaic methylammonium lead iodide. The journal of physical chemistry letters7(17), 3458-3466.

Reviewer 3 Report

Authors have fabricated 2D perovskite using PbI2 - 2-aminoethyl methacrylate hydrochloride and Pb(NO3)2 - 2-aminoethyl methacrylate hydrochloride. Then characterized them using different physical analysis techniques at different time interval. Also they have used this 2D perovskite as a NH3 gas sensor. Which is interesting concept to use a gas sensor. But the manuscript title does not reflect the manuscript. The title should indicate the use 2D perovskite as a gas sensor.

  1. How did you confirm the structure of 2D perovskite "H2C=C(CH3)CO2CH2CH2NHPbCl3"? Why not "H2C=C(CH3)CO2CH2CH2NHPbClxIx-1"? Please provide evidence.
  2. Also both "I" and "N" produced same structure of perovskite "H2C=C(CH3)CO2CH2CH2NHPbCl3". How it is possible?
  3. Most important figure A1 is in supplementary. It should be in main manuscript.
  4. Figure 3 and description of Figure 3 does not match properly. Figure 3 should be modified according to the description. Especially in the first step substrate should be dipped in to mixed I-AEMP or N-AEMP according to the description.
  5. Authors dipped the substrate in the solution for 30 min then spin coating in two steps. But why? Why not using drop of solution and then spin coating? Please explain properly.
  6. After film formation annealed at 200oC for 1 hour, isn't it too high temp and too long time for organic based perovskite? Also what was the annealing environment? What is the difference between as deposited and annealed one?
  7. XRD showing peak near 24.8, which was identified as <101> (Figure A4) but Figure 8 and 9 in different 2Θ positions. Which one is correct and how can you identify as perovskite film with only one weak peak. Also it was matched with inorganic perovskite. Please provide JCPDS card number and confirm your film as perovskite.
  8. Samples kept in S and V condition showed higher degradation than RT condition according to the figure 5. Can you explain how it is possible?
  9. To understand film quality based on morphology and topography at least FE-SEM or SEM image is required. Please provide them.

Author Response

Comments and Suggestions for Authors

Actions by Authors

Reviewers 3

Authors have fabricated 2D perovskite using PbI2 - 2-aminoethyl methacrylate hydrochloride and Pb(NO3)2 - 2-aminoethyl methacrylate hydrochloride. Then characterized them using different physical analysis techniques at different time interval. Also they have used this 2D perovskite as a NH3 gas sensor. Which is interesting concept to use a gas sensor. But the manuscript title does not reflect the manuscript. The title should indicate the use 2D perovskite as a gas sensor.

Effect of Stability of Two Dimensional (2D) Aminoethyl Methacrylate Perovskite using Lead-based Materials in Exposed Ammonia Gas

was changed to

Effect of Stability of Two Dimensional (2D) Aminoethyl Methacrylate Perovskite using Lead-based Materials for Ammonia Gas Sensor Application

How did you confirm the structure of 2D perovskite "H2C=C(CH3)CO2CH2CH2NHPbCl3"? Why not "H2C=C(CH3)CO2CH2CH2NHPbClxIx-1"? Please provide evidence.

·     We have referred to literature and we have revised the structure of 2D which based on (RNH3)2MnX3n+1 with n values is 1.

·     Hence, we have replaced the figure in Figure 2 with chemical equation of both materials. (Page no. 3)

Also both "I" and "N" produced same structure of perovskite "H2C=C(CH3)CO2CH2CH2NHPbCl3". How it is possible?

We have made correction as shown in Figure 2 (Page no. 3).

Most important figure A1 is in supplementary. It should be in main manuscript.

Figure A1 in supplementary was transfer to main manuscript as Figure 3, as suggested. Thus, there are some changes on the respective Figure no and captions:

Figure 3 changed to Figure 4

Figure 4 changed to Figure 5

Figure 5 changed to Figure 6

Figure 6 changed to Figure 7

Figure 7 changed to Figure 8

Figure 8 changed to Figure 9

Figure 9 changed to Figure 10

Figure 10 changed to Figure 11

Figure 11 changed to Figure 12

Figure 12 changed to Figure 13

Figure 13 changed to Figure 14

Figure 3 and description of Figure 3 does not match properly. Figure 3 should be modified according to the description. Especially in the first step substrate should be dipped in to mixed I-AEMP or N-AEMP according to the description.

The description of Figure 3 was changed to suitable sentences. Please accept this changes.

Authors dipped the substrate in the solution for 30 min then spin coating in two steps. But why? Why not using drop of solution and then spin coating? Please explain properly.

This is due to after some trials and errors, it was found that this method could produce better surface coating and high quality of thin film.

After film formation annealed at 200°C for 1 hour, isn't it too high temp and too long time for organic based perovskite? Also what was the annealing environment? What is the difference between as deposited and annealed one?

After some trials and errors on using different temperatures, this temperature is found an optimum temperature for fully conversion of starting materials to perovskite (dark brown colour). The choice of 200°C also due to the use of ethylene glycol solvents which has  boiling point around 197°C.

XRD showing peak near 24.8, which was identified as <101> (Figure A4) but Figure 8 and 9 in different 2Θ positions. Which one is correct and how can you identify as perovskite film with only one weak peak. Also it was matched with inorganic perovskite. Please provide JCPDS card number and confirm your film as perovskite.

Actually Figure A4  is the same spectrum of D1 at RT of Figure 8 (current draft: Figure 9) for I-AMP. <101> that represent to perovskite peak is in different location of 2Ɵ because Figure 9  (current draft: Figure 10) is for N-AMP. We have confirmed the location based on previous perovskite reports (as in page no. 9) and after checking final spectrum with starting materials (Figures A3).

Samples kept in S and V condition showed higher degradation than RT condition according to the figure 5. Can you explain how it is possible?

Based on overall analysis, not just on surface morphology analysis, the results showed that The decrease in crystallite size, crystallinity and band gap with time confirm the decrease in the stability of I-AMP materials. Whilst for N-AMP, the results show otherwise—crystallite size, crystallinity and band gap energy revealed increasing trends over time, hinting the superior stability of the material. Whilst, for I-AMP, incomplete coverage of perovskite materials on the surface can be seen especially under RT and surface morphology changes were identified over time. In contrast, N-AMP material shows almost fully coverage of perovskite materials under all conditions.

To understand film quality based on morphology and topography at least FE-SEM or SEM image is required. Please provide them.

It is out of our research scope for this study to analyse the samples using FESEM/SEM. However,  there is a plan to further analyse using this equipment for better observation.

Reviewer 4 Report

Adli reported an interesting work about the stability of perovskite under NH3 gas. The stability of perovskite is a currently very hot area to explore, and this work provide a good idea to study the effect on the stability. I think this work is worthwhile to publish, however, a couple of experimental and writing flaws are presented in the work. Hence, a major revision must be made before publishing it.

  1. The drawing of Figure 2 is confusing. It cannot really show the crystal structure of the perovskite. I suggest the authors delete the figure 2, or draw the structure model like the Figure 1.
  2. In the figure 1 caption, the authors should note what are A,B and X.
  3. The fabrication of 2D-aminoethyl methacrylate perovskites using PbI2 is confusing to me. Will the I also doped inside the crystal structures? Do authors have any evidence?
  4. For some words like room temperature (RT), silica (S), D1, and vacuum (V), I suggest the authors do not use the abbreviation, as it is hard to read, especially use only one letter.
  5. The scale bar must be provided in Figure 5, instead of the magnification.
  6. Page 6, “Islands of perovskite materials became enhanced resulting to poor coverage on the substrate, highlighting the degradation of perovskite that occurred”. I do not quite understand what the authors what to express. The island structure is desired? Why does poor coverage on the substrate lead to that?
  7. I think the introduction about the advantages of perovskites materials should be stressed. A couple of more sentences and references should be added. They are not only used for optoelectronic and photonic device applications, but also has been used for photocatalysis and laser cooling and others. The detailed devices such as solar cell and LED could be listed instead just using “ optoelectronic and photonic device”. More related papers in photocatalysis (ACS Macro Lett. 2020, 9, 5, 725–730; J. Am. Chem. Soc. 2019, 141, 2, 733–738), solar cell, LED and others should be cited.
  8. Figure 6, why does the λmax of two perovskites are different. They should have the same composition and crystal structures. The difference is just the precursors.
  9. I do not quite agree with the conclusion the authors made in Figure 6, about the better stability of N-AMP. Although the band gap increased, the UV absorption intensity decreased a lot, suggesting the possible degradation, or at least the lost of ability in photon absorption.
  10. For the XRD of I-AMP, the authors can only observe a perovskite peak (101). This is very weird as other peaks such as (111) should also be observed. are you sure this is the peak of perovskite rather than some impurities?
  11. Figure 8, the authors said (101) is located at 24.8. But it is clearly over 25 degree.
  12. Figure 8, in other storage conditions, (101) has a red shift. The authors said it is due to moisture-induced decomposition. Any reference support? The shift is 4 degree and is the way too large. Also under vacuum condition there should not be any moisture.

Author Response

Comments and Suggestions for Authors

Actions by Authors

Reviewers 4

The drawing of Figure 2 is confusing. It cannot really show the crystal structure of the perovskite. I suggest the authors delete the figure 2, or draw the structure model like the Figure 1.

•             Chemical structure in Figure 2 was replaced with chemical equation for both peroskite.

•             As we revised, there are two compounds were identified which are (H2C=C(CH3)CO2CH2CH2NH3)2PbI2Cl2 (I-AMP) and (H2C=C(CH3)CO2CH2CH2NH3)2Pb(NO3)2Cl2 (N-AMP)

In the figure 1 caption, the authors should note what are A,B and X.

The caption of Figure 1 is revised based on suggestion.

The fabrication of 2D-aminoethyl methacrylate perovskites using PbI2 is confusing to me. Will the I also doped inside the crystal structures? Do authors have any evidence?

·                 Based on Figure 4 (current draft), the synthesis of AMP was conducted using 2-Aminoethyl methacrylate hydrochloride (AEMA) and lead-based materials (I or N).

·                 The mixed solution of AMP (one for I-AMP, another for N-AMP) was then used directly, in which the TiO2 fim was put into the solution.

For some words like room temperature (RT), silica (S), D1, and vacuum (V), I suggest the authors do not use the abbreviation, as it is hard to read, especially use only one letter.

We have revised the overall content related to the comment.

The scale bar must be provided in Figure 5, instead of the magnification.

In this case, we use Optical Microscope instrument in which only provided scale bar of image that represent image magnification.

Page 6, “Islands of perovskite materials became enhanced resulting to poor coverage on the substrate, highlighting the degradation of perovskite that occurred”. I do not quite understand what the authors what to express. The island structure is desired? Why does poor coverage on the substrate lead to that?

The statement was changed from:

Islands of perovskite materials became enhanced resulting to poor coverage on the substrate, highlighting the degradation of perovskite that occurred.

to

Poor coverage on the substrate was found as the formation of the islands was observed, hinting the degradation of perovskite was occurred under RT and S conditions (page no. 6)

I think the introduction about the advantages of perovskites materials should be stressed. A couple of more sentences and references should be added. They are not only used for optoelectronic and photonic device applications, but also has been used for photocatalysis and laser cooling and others. The detailed devices such as solar cell and LED could be listed instead just using “ optoelectronic and photonic device”. More related papers in photocatalysis (ACS Macro Lett. 2020, 9, 5, 725–730; J. Am. Chem. Soc. 2019, 141, 2, 733–738), solar cell, LED and others should be cited.

Additional sentences and references were added for introduction part as suggested:

·        Currently, perovskite materials also have been explored to be used for laser cooling and photocatalysis (Zhu et al., 2020; Zhu et al., 2019).

·        Zhu, Y., Liu, Y., Miller, K. A., Zhu, H., & Egap, E. (2020). Lead halide perovskite nanocrystals as photocatalysts for PET-RAFT polymerization under visible and near-infrared irradiation. ACS Macro Letters9(5), 725-730.

·        Zhu, X., Lin, Y., Sun, Y., Beard, M. C., & Yan, Y. (2019). Lead-halide perovskites for photocatalytic α-alkylation of aldehydes. Journal of the American Chemical Society141(2), 733-738.

·        Due to the growing interest in perovskite materials, variety of application have been explored for solar cell, light-emitting devices, semiconductor device, transistor, waste management applications as well as gas sensor applications [11-14].

Figure 6, why does the λmax of two perovskites are different. They should have the same composition and crystal structures. The difference is just the precursors.

After some revision and changes have been made on Figure 2, hence there are different λmax for each material that has different composition and crystal structure.

I do not quite agree with the conclusion the authors made in Figure 6, about the better stability of N-AMP. Although the band gap increased, the UV absorption intensity decreased a lot, suggesting the possible degradation, or at least the lost of ability in photon absorption

We have revised the draft. Although the bandgap energy correlate to crystallite size, which based on some references can relate to the stability of material.

Few related references were cited throughout the articles such as:

·        Change in crystallite size can be realized due to the increase of dislocation density and micro-strain. Guo et al stated that perovskite degradation causes phase segregation, morphological deformation, and lattice shrinkage, which could contribute to crystallite size reduction [104].

·        The decline in crystallite size of I-AMP may also be due to decomposition of perovskite into PbI2 that leads to point and interstitial defects, and/or vacancy into the perovskite lattice, that causes dislocation of atoms in the structure [106].

·        The decrease in crystallite size depicts gradual degradation of perovskite material [70].

However, we also indentifed that the UV absorption intensity of N-AMP was decreased a lot, that might be due to the condition of the mixed solutions that were kept until D20. From the naked eyes, that solution formed some precipitates and non-homogeneous solution. However, we are taking noted on this matter for further analysis in depth.

For the XRD of I-AMP, the authors can only observe a perovskite peak (101). This is very weird as other peaks such as (111) should also be observed. are you sure this is the peak of perovskite rather than some impurities?

We have confirmed the perovskite location based on previous perovskite reports (as in page no. 9) and after checking final spectrum with starting materials (Figures A3 and A4). Based on physical observation on the colour changes of thin film from yellow to typical brown colour of perovskite also -suggesting that the formation of perovskite was occurred.

Figure 8, the authors said (101) is located at 24.8. But it is clearly over 25 degree.

Excuse the error, we have modified the content. The perovskite peak (101) is located at 25.35° (Page no. 10)

Figure 8, in other storage conditions, (101) has a red shift. The authors said it is due to moisture-induced decomposition. Any reference support? The shift is 4 degree and is the way too large. Also under vacuum condition there should not be any moisture.

We have cited relevant paper to support, please accept this justification.

Hofstetter, Y. J., García-Benito, I., Paulus, F., Orlandi, S., Grancini, G., & Vaynzof, Y. (2020). Vacuum-induced degradation of 2D perovskites. Frontiers in chemistry, 66.

Round 2

Reviewer 3 Report

Thanks authors for their reply and modification of the manuscript.

Author Response

Thank you for suggestion and comments for our manuscript.

Reviewer 4 Report

The revised version is much better. However, some questions need to be further addressed.

  1. The absorption decreased of N-AMP is still not very clear. I-AMP in D20 is still similar. As the stability of N-AMP is stronger than I-AMP as the authors claimed, why there is still a larger decrease in N-AMP? Some more discussion should be added, even if the experiment evidence will not be provided in this work.
  2. For the XRD of I-AMP, the authors still did not directly answer my question: why did you only observe one peak? Due to the limited  crystallinity? Or other reason? 
  3. I would suggest the authors add one or two sentences' discussion about the stability of  perovskite. The authors only discuss the structure influence on perovskite. However, surface coating with polymer, COF or SiO2 could also enhance the stability. Some papers could be cited (ACS Materials Lett. 2022, 4, 3, 464–471;J. Am. Chem. Soc. 2016, 138 (18), 5749−5752)
  4. Following the question 3, the authors could stress a little about why does two D perovskite is a better strategy to enhance stability than surface coating. For examples, it will not influence the surface activity sites, not changing the optical properties and others.

Author Response

COVER LETTER FOR EDITORS

Comments and Suggestions for Authors

Actions by Authors

Reviewers 4

The absorption decreased of N-AMP is still not very clear. I-AMP in D20 is still similar. As the stability of N-AMP is stronger than I-AMP as the authors claimed, why there is still a larger decrease in N-AMP? Some more discussion should be added, even if the experiment evidence will not be provided in this work.

For your information, UV-vis analysis was conducted using mixed solution of I-AMP and N-AMP as samples. Based on Figure 7, it can be seen that the absorption of I-AMP kept almost the same up until D20, but I-AMP experience decreased absorption for the case of D20. One factor that influences the absorbance of a sample is the concentration (c) of the solution.  The decreased absorption of N-AMP at D20 can be explained due to the quality of mixed solution at D20, which we observed small precipitates formed at the bottom of tube. In this condition, when we test the solution for UV-vis, the formed  precipitate might block the absorption of UV and eventually caused poor UV absoption.

Figure shows the comparison of mixed solution of I-AMP and N-AMP at D20. It can observed that I-AMP was in homogeneous solution whilst, N-AMP otherwise.

I-AMP

N-AMP

However, when we calculate the bandgap energy (Figure 8), a slight decrease in band gap energy over time for the case I-AMP and the band gap energy of N-AMP trend shows otherwise. The stability was observed based on band gap energy and cystallite size as the crystallite size of the materials are predominantly influenced by the band gap energy [77,83]. The optical band gap energy is determined based on Tauc equation where the optical band gap was obtained from the respective tangents to x-axis when y is equal to 0, which not related to absorption intensity.

From Structural and Crystallinity analysis, the change in crystallite size can be realized due to the increase of dislocation density and micro-strain. Guo et al stated that perovskite degradation causes phase segregation, morphological deformation, and lattice shrinkage, which could contribute to crystallite size reduction [113]. The decline in crystallite size of I-AMP may also be due to decomposition of perovskite into PbI2 that leads to point and interstitial defects, and/or vacancy into the perovskite lattice, that causes dislocation of atoms in the structure [115]. In contrast, crystallite size of N-AMP increased over time under all conditions, which tallies with the trend of band gap energy.

For the XRD of I-AMP, the authors still did not directly answer my question: why did you only observe one peak? Due to the limited  crystallinity? Or other reason?

Based on the spectrum, yes we only can observe one peak perovskite can be observed.

 From Figure 9, it can be seen that peak perovskite of I-AMP was observed at around 25.35° in RT condition. Other peaks were belong as PbI2 and TiO2 (based on reference). However, for S and V conditions, the spectrum become amorphous and it causes the peak become broaden. Hence, the peak perovskite of I-AMP only can be found at around 28.5° - 29.5°.

For N-AMP in Figure 10, the peak at (111) is belong to Pb(NO3)2 based on the spectrum of starting material and previous studies.

Thus, both N-AMP and I-AMP exhibited only one peak perovskite of (101). We kind of agree with reviewer that the materials formed are in limited crystallinity, especially because AEMA is in amorphous (as in Figure A3).

However, few references were added for supporting the discussion.

New References:

·     Lee, J.W.; Dai, Z.; Han, T.H.; Choi, C.; Chang, S.Y.; Lee, S.J.; Yang, Y. 2D perovskite stabilized phase-pure formamidinium perovskite solar cells. Natur. Communi. 2018, 9, 1-10.

·     Yao, K.; Wang, X.; Li, F.; Zhou, L. Mixed perovskite based on methyl-ammonium and polymeric-ammonium for stable and reproducible solar cells. Chem. Communi. 2015, 51, 15430-15433.

I would suggest the authors add one or two sentences' discussion about the stability of  perovskite. The authors only discuss the structure influence on perovskite. However, surface coating with polymer, COF or SiO2 could also enhance the stability. Some papers could be cited (ACS Materials Lett. 2022, 4, 3, 464–471;J. Am. Chem. Soc. 2016, 138 (18), 5749−5752)

We agreed with the suggestion by the reviewer. Hence, new sentences and references were added as follow.

Page no 3:

However, previous studies have reported that titanium dioxide layers preserved adequate porosity with a small amount of physisorbered water, which led to an increase in perovskite conversion from starting materials (Lee et al., 2014; Burschka et al., 2013). In addition, surface coating with polymer, COFs or SiO2 could also enhance the stability (Zhu et al., 2022; Huang et al., 2016). According to Zhu et al, the covalent organic frameworks (PNCs–COFs) demonstrated good water processability that originates from the hydrophobic COFs shell. Furthermore, the COFs coating enables the formation of PNCs–COFs heterojunction, which facilitates the generation and transportation of photoinduced charge carriers (Zhu et al., 2022). Huang et al also have reported that the inorganic material coatings of SiO2 are preferred for coupling with perovskites to improve their stability, whereas the conventional SiO2 formation method is unsuitable as it requires water (Huang et al., 2016). The structural stability of perovskite also is characterized by the absence of polymorphism and the capacity to remain stable in a particular crystalline period under very broad circumstances, such as heat and pressure (Berhe et al., 2016).

References:

·     Burschka, J., Pellet, N., Moon, S. J., Humphry-Baker, R., Gao, P., Nazeeruddin, M. K., & Grätzel, M. (2013). Sequential deposition as a route to high-performance perovskite-sensitized solar cells. Nature, 499(7458), 316-319.

·     Berhe, T. A., Su, W. N., Chen, C. H., Pan, C. J., Cheng, J. H., Chen, H. M., ... & Hwang, B. J. (2016). Organometal halide perovskite solar cells: degradation and stability. Energy & Environmental Science9(2), 323-356.

·     Lee, J. W., Lee, T. Y., Yoo, P. J., Grätzel, M., Mhaisalkar, S., & Park, N. G. (2014). Rutile TiO 2-based perovskite solar cells. Journal of Materials Chemistry A2(24), 9251-9259.

·     Zhu, Y., Liu, Y., Ai, Q., Gao, G., Yuan, L., Fang, Q., ... & Lou, J. (2022). In situ synthesis of lead-free halide perovskite–COF nanocomposites as photocatalysts for photoinduced polymerization in both organic and aqueous phases. ACS Materials Letters4(3), 464-471.

·     Huang, S., Li, Z., Kong, L., Zhu, N., Shan, A., & Li, L. (2016). Enhancing the stability of CH3NH3PbBr3 quantum dots by embedding in silica spheres derived from tetramethyl orthosilicate in “waterless” toluene. Journal of the American Chemical Society138(18), 5749-5752.

Following the question 3, the authors could stress a little about why does two D perovskite is a better strategy to enhance stability than surface coating. For examples, it will not influence the surface activity sites, not changing the optical properties and others.

We agreed with the suggestion by the reviewer. Hence, new sentence and references were added as follow.

Page no 3:

2D perovskite becomes more stable than 3D counterpart due to relaxation of the hydrogen bonds at the surface that contributed the degradation speed of 2D perovskites is much slower than their 3D materials [42-43]. By focusing on the material itself, it will not influence on the surface activity sites, the optical properties of materials for gas sensor and flexible for the device fabrication.

References:

·     Kim, M., Alfano, A., Perotto, G., Serri, M., Dengo, N., Mezzetti, A., ... & Lamberti, F. (2021). Moisture resistance in perovskite solar cells attributed to a water-splitting layer. Communications Materials2(1), 1-12.

·     Yang, Y., Gao, F., Gao, S., & Wei, S. H. (2018). Origin of the stability of two-dimensional perovskites: a first-principles study. Journal of Materials Chemistry A6(30), 14949-14955.
